# Appetite-Suppressing and Satiety-Increasing Bioactive Phytochemicals: A Systematic Review

**DOI:** 10.3390/nu11092238

**Published:** 2019-09-17

**Authors:** Johann Stuby, Isaac Gravestock, Evelyn Wolfram, Giuseppe Pichierri, Johann Steurer, Jakob M. Burgstaller

**Affiliations:** 1Horten Centre for Patient Oriented Research and Knowledge Transfer, University of Zurich, 8032 Zurich, Switzerland; isaac.gravestock@usz.ch (I.G.); giuseppe.pichierri@usz.ch (G.P.); johann.steurer@usz.ch (J.S.); jakob.burgstaller@usz.ch (J.M.B.); 2ZHAW Life Sciences und Facility Management, Phytopharmacy & Natural Product Research Group, 8820 Waedenswil, Switzerland; wola@zhaw.ch

**Keywords:** bioactive phytochemicals, plant extracts, appetite, hunger, satiety, fullness

## Abstract

The prevalence of obesity is increasing worldwide. Bioactive phytochemicals in food supplements are a trending approach to facilitate dieting and to improve patients’ adherence to reducing food and caloric intake. The aim of this systematic review was to assess efficacy and safety of the most commonly used bioactive phytochemicals with appetite/hunger-suppressing and/or satiety/fullness-increasing properties. To be eligible, studies needed to have included at least 10 patients per group aged 18 years or older with no serious health problems except for overweight or obesity. Of those studies, 32 met the inclusion criteria, in which 27 different plants were tested alone or as a combination, regarding their efficacy in suppressing appetite/hunger and/or increasing satiety/fullness. The plant extracts most tested were derived from *Camellia sinensis* (green tea), *Capsicum annuum*, and *Coffea* species. None of the plant extracts tested in several trials showed a consistent positive treatment effect. Furthermore, only a few adverse events were reported, but none serious. The findings revealed mostly inconclusive evidence that the tested bioactive phytochemicals are effective in suppressing appetite/hunger and/or increasing satiety/fullness. More systematic and high quality clinical studies are necessary to determine the benefits and safety of phytochemical complementary remedies for dampening the feeling of hunger during dieting.

## 1. Introduction

Overweight and obesity are becoming more frequent worldwide. According to the World Health Organization (WHO), the global prevalence of obesity has nearly tripled since 1975. More than 1.9 billion adults are presently overweight (BMI > 25 kg/m^2^), of which over 650 million are considered obese (BMI > 30 kg/m^2^) [1]. In Europe, approximately 23% of women and 20% of men are obese [2]. Overweight is associated with an increased risk of a number of chronic diseases, such as type 2 diabetes, cardiovascular diseases, degenerative arthritis, and several malignancies (e.g., liver and colon) [1,3,4]. Therefore, the prevention and therapy of overweight and obesity are of clinical importance.

Many different approaches exist to reduce weight, including lifestyle changes, pharmacotherapy, and bariatric surgery [5,6,7]. When lifestyle changes, such as increased physical activity and reduced energy intake (“move more, eat less”), are ineffective and body mass index (BMI) is over 30 kg/m^2^ (without obesity-related comorbidities, or over 27 kg/m^2^ with obesity-related comorbidities), pharmacotherapy is often recommended [8,9]. Several approved pharmaceutical agents are on the market, including orlistat, amfepramone, liraglutide, and naltrexone/bupruprion [10]. The last resort for the treatment of weight loss remains bariatric surgery, which is reserved for people with extreme weight problems (unsuccessfully completed weight loss programs, obesity-related comorbidities, and BMI > 40 kg/m^2^ or BMI > 35 kg/m^2^) [11]. The serious side effects of pharmacotherapy (e.g., anxiety, insomnia, hepatotoxicity, nephrotoxicity, primary pulmonary hypertension [9,10,12]) and of bariatric surgery (e.g., infection, deep vein thrombosis, acute pancreatitis, hernia [13,14]), as well as the fact that the majority of obese patients do not fulfill the criteria for those interventions, are reasons why there is an urgent need for alternative treatments [7,10,11,15].

In order to facilitate weight loss, some bioactive phytochemicals, derived from plants that are not specifically known as nutrient compounds, have recently gained considerable popularity [16,17]. Besides enhancing the rate of metabolism, these potential naturally derived food supplements may also act to suppress appetite [18]. The mechanisms of action of this effect are still not fully understood [19,20,21]. Possible targets are, among others, gut-derived hormones (such as GLP-1 or ghrelin) [20,22] or receptors of the neuronal system (such as TRPV1) [23]. The appetite-suppressing property could be useful as a complementary agent for a variety of different diets, in which a nagging feeling of hunger is often one of the crucial problems, especially in the beginning [24,25]. Various natural remedies that may suppress the feeling of appetite have been studied. Herbal drugs (e.g., green tea or yerba mate) and their extracts, fractions, and even isolates (e.g., catechins) are claimed to be useful for weight loss [8,26,27,28,29,30]. Safety and clinical efficacy of those supplements often remain unclear, since such studies in food are not required by market regulatory authorities as in the pharmaceutical sector. There is a possibility that these substances may even interact with prescription drugs and can possibly cause adverse events [17,31,32,33], which may not be known since phytochemical supplements do not require a pharmacovigilance monitoring as do market authorized phytotherapeutic remedies [34]. Thus, there is a need for scientific evidence of product-specific quality, efficacy, and safety of bioactive phytochemicals used in weight-loss targeted treatments.

The aim of this study was to review systematically the literature on the assessment of efficacy and safety of bioactive phytochemicals (water or solvent extracts, fractions, or isolates) with appetite/hunger-suppressing and/or satiety/fullness-increasing properties.

## 2. Materials and Methods

This systematic review is based on the “Preferred Reporting Items for Systematic Reviews and Meta-Analyses” (PRISMA) [35].

### 2.1. Literature Search

In March 2019, an experienced librarian conducted a systematic literature search at Careum library of the University of Zurich in the following five electronic databases: Cochrane Library, Embase, Ovid MEDLINE, PsycINFO, and Scopus. The following search terms, among others, were used: “phytochemical”, “phytonutrient”, “phytoconstituent”, “bioactive ingredient”, “nutraceutical”, “dietary supplement”, “medicinal herbs and plants”, “plant extracts”, “plant medicinal product”, “appetite depressing drugs”, “phenol”, “polyphenol”, “terpene”, “carotenoid”, “alkaloid”, “appetite”, “hunger”, “satiety”, “fullness”, “satiation”, “suppress”, “control”, “regulation”, “obesity”, “overweight”, “weight loss”, “weight management”, “energy intake”, and “energy balance”. One detailed search strategy with all used terms is shown in Appendix A. No restrictions concerning type of study or publication date were applied.

### 2.2. Definition of Bioactive Phytochemicals

We consider bioactive phytochemicals to be secondary metabolite compounds derived from plants that are non-nutrients and have an effect on the human organism according to the definitions of Liu [36] and Huang et al. [37] complemented by Saldanha et al. [38] for the regulatory perspective on the specification of a botanically derived health beneficial “dietary ingredient”. In analogy to phytotherapy as a complementary remedy in the pharmaceutical treatment of patients, such bioactive phytochemicals can be applied as a complementary food supplement in therapeutic diets.

### 2.3. Inclusion and Exclusion Criteria

Studies included in this review were randomized controlled trials (RCTs) that compared the appetite/hunger-suppressing or satiety/fullness-increasing effect of an orally administered plant extract containing as main compounds at least one or more bioactive phytochemicals compared to a placebo. To be eligible, RCTs had to include at least 10 patients per group. In the case of crossover RCTs (Co-RCTs), each test person underwent both, intervention as well as control treatment. This criterion was based on the smallest sample size that was justified by a power analysis from literature previously known to us. To be enrolled, the test persons had to be aged 18 years or older with no serious health problems except for overweight (BMI > 25 kg/m^2^) or obesity (BMI > 30 kg/m^2^). A valid and reliable instrument (e.g., visual analogue scale (VAS)) had to be used to quantify appetite, hunger, satiety, and/or fullness. We focused on studies using subjective parameters (patient-reported outcome measures), whereas objective endocrine markers were not considered. The evidence for a role of appetite-related peptides is far from clear. At this stage, no composite peptide measure exists that is similar to a subjective appetite rating [39].

Studies were excluded if the plant extract tested consisted mainly of macronutrients (e.g., carbohydrates, proteins, fats), conventional foodstuff (e.g., potato, rice, wheat products), or pharmaceutical medications, or if it had an addictive potential (e.g., nicotine or marihuana). We included studies investigating caffeine because there is an ongoing debate in the literature about whether caffeine has an addictive potential [40,41,42,43]. Herbal teas, when the herbal substance is monographed in Ph,Eur [44] or in an official monograph such as one issued by the Committee on Herbal Medicinal Products (HMPC) [45], were not regarded as conventional foodstuff, since they are also considered in phytotherapy as an active remedy. In addition, studies were excluded if the identity and content of the bioactive phytochemicals in the tested plant extract were not specified to ensure reproducibility or if there existed toxicological concerns. Trials that included patients with anorexia nervosa or bulimia nervosa or pregnant women were not considered. Furthermore, studies that used animal models as well as abstracts and study protocols were excluded. When the same study was included in several publications without change in treatment, outcome, or follow-up, the most recent publication was chosen and missing information was added from previous publications.

### 2.4. Study Selection

Two reviewers independently performed the initial screening of the references for relevance by title and abstract. Following this, full text analysis for eligibility was conducted. Any disagreements were discussed and solved by consensus or third-party arbitration.

### 2.5. Quality Assessment

To review the quality of the included RCTs, the checklist of the Scottish Intercollegiate Guidelines Network (SIGN) for RCTs was used [46]: addressing an appropriate and clearly focused question, proper conduction of randomization, allocation concealment, blinding, similarity of treatment and control groups at baseline, measure of the outcomes in a reliable way, percentage of dropout, use of intention-to-treat analysis, and minimizing of bias. If the majority of the criteria were met (if at most one criterion was answered with a “no” or “can’t say”), the study assessed a high rating. Acceptable quality was provided if most criteria were met with few flaws (if at most two criteria were answered with a “no” or “can’t say”). Studies were rated low quality if either most criteria were not fulfilled or significant flaws in key aspects of the study design were present (if at most four criteria were answered with a “no” or “can’t say”). If almost none of the SIGN quality criteria were met (if more than four criteria were answered with a “no” or “can’t say”), the study was considered unacceptable and therefore was rejected. For Co-RCTs, criterion 1.5 was not applicable and therefore not considered for the final rating.

### 2.6. Data Extraction

For each eligible study, the full text was obtained and baseline characteristics of the subjects (such as age, gender, BMI, waist and hip circumference) were extracted into a Microsoft Excel^®^ (2016) worksheet. In addition, outcomes of interest, standard information of the trials (aim, conclusion, study product, inclusion and exclusion criteria, etc.) and adverse events were recorded.

### 2.7. Outcomes

The primary outcome of interest of this systematic review was the decrease of appetite or hunger, and/or the increase of satiety or fullness, after consumption of a bioactive phytochemical, compared to a control. 

As secondary outcome of interest, safety (occurrence of adverse events) of the plant extracts was assessed.

### 2.8. Statistics

Due to the heterogeneity of the included studies (different phytochemicals, appetite scales, follow-up times etc.), no meta-analysis was performed. Therefore, descriptive analysis and comparative analysis were used to present the study results. The most important findings are summarized in tables. 

In the manuscript, the study results were presented with “0” and “+”. Studies that failed to show a statistically significant difference (*p* > 0.05) between intervention and placebo were rated with “0”. If the plant extract tested achieved a significant difference compared to placebo (*p* < 0.05) in at least one of the four outcomes (appetite, hunger, fullness, or satiety), it was assessed as “+”. 

A more detailed presentation of the study results can be found in the Appendix A. If no significant treatment effect was found (*p* > 0.05), studies were rated with “0”. In the case of a significant treatment effect (*p* < 0.05) in favor of the placebo group, the trial achieved a “−”. A significant treatment effect (*p* < 0.05) in favor of the intervention group in at least one follow-up comparison was rated with “+” and a significant treatment effect (*p* < 0.05) in favor of the intervention group in all follow-up comparisons with “++”.

## 3. Results

### 3.1. Study Selection

The systematic literature search identified 5510 potentially eligible studies. Of these, 2124 were excluded after manual and automatic deduplication. A bibliography screening revealed one additional publication, leading to 3387 articles. After screening for title and abstract, 3344 studies were eliminated. The full texts of the remaining 43 papers were reviewed with regard to the inclusion and exclusion criteria and to the quality assessment (SIGN) [46]. Finally, 32 studies met the inclusion criteria and were of acceptable quality. The systematic flowchart, including the main reasons for exclusion, is displayed in Figure 1.

### 3.2. Study Overview

The baseline characteristics of the participants of all included studies are summarized in Appendix A. Out of 32 reviewed studies, 17 were parallel-designed randomized controlled trials (RCT,) and 15 were crossover-designed randomized controlled trials (Co-RCTs). All of them were conducted between 2003 and 2018. Overall, extracts of 27 different plants were tested alone or as a combination (Table 1). The most common plant extracts examined alone were derived from *Camellia sinensis* (green tea) in nine trials, *Capsicum annuum* in five trials, and *Coffea* species in four trials. Further plant extracts used alone or in combination with other plant extracts, and the corresponding number of studies, are listed in Table 1. In the following text, the Latin species names are used without suffix. For more information about the plant extracts (plant of origin, bioactive phytochemical, placebo, etc.) and the nutritional regimens see Appendix A).

Taking all studies together, 1481 participants were included (median 43 participants, range 11 to 138) with a mean age of 30.4 years and a mean percentage of women of 64%. Four trials (13%) only included men, six trials (19%) only women. In 13 trials (41%), the participants were on average normal weighted (BMI < 25 kg/m^2^), in 12 trials (38%) they were on average overweight (BMI ≥ 25 to < 30 kg/m^2^), and in 4 trials (13%) they were on average obese (BMI ≥ 30 kg/m^2^). The mean BMI was 26.1 kg/m^2^ and the mean waist–hip ratio (WHR) was 0.85 (according to the WHO, a WHR of ≥0.90 for men and ≥0.85 for women is defined as abdominal obesity [47]). A total of 23 studies (72%) reported at least one of the primary outcomes of interest (appetite, hunger, satiety, and/or fullness) using the visual analogue scale (VAS, 0 to 100 mm), four studies (13%) using the Haber scale (−10 to +10), and the remaining six studies using in total six different scales. One study used two different scales. While most RCTs recorded the outcomes over several days (mean 85 days, range 21 to 365 days), crossover RCTs usually recorded them over several hours (mean 5 h, range 1 to 36 h) on just one or two days, respectively.

### 3.3. Primary Outcome

The outcomes regarding a decrease of appetite/hunger and/or an increase of satiety/fullness for plant extracts examined in several trials are summarized in Table 2, rated with “+” or “0”. None of these substances showed a consistent significant difference between the active substance and placebo. Detailed results of all analyzed studies are illustrated in Appendix A (rating ranging between “++”, “+”, “0”, and “−“).

#### 3.3.1. *Camellia sinensis* (Green Tea)

Green tea (made from leaves of *Camellia sinensis* L. Kuntze) contains two major active ingredients, catechin polyphenols and caffeine [77,78]. The polyphenols include, among others, epicatechin (EC), epicatechin gallate (ECG), epigallocatechin (EGC), and epigallocatechin gallate (EGCG) [53,79,80].

In a short-term crossover RCT of Fernandes et al. [51], an intake of 752 mg EGCG led to a significant augmentation of fullness after 90 min compared to placebo, whereas there was no effect on hunger. Similar effects where shown in a trial of Josic et al. [52], in which 32.4 mg EGCG resulted in increased fullness and satiety, but not in significantly reduced hunger.

Most of the long-term studies (ranging from 21 to 365 days) investigating the appetite/hunger-decreasing and/or satiety/fullness-increasing properties of catechins failed to show a positive treatment effect. In an RCT from Auvichayapat et al. [48] using 100.7 mg EGCG daily for three months, the satiety score was not significantly different compared to the placebo group. The RCTs from Diepvens et al. [49], Mangine et al. 2012 [55], and Westerterp-Plantenga et al. [54], which investigated the impact of 596 mg EGCG, 105 mg EGCG, and 270 mg EGCG daily, respectively, came to the same result.

Opposite effects were found by Reinbach et al. [53]. A daily intake of 1796 mg catechins (content of EGCG not reported) for three weeks was able to increase fullness (but not satiety) and to decrease appetite. This result is in line with a study from Rondanelli et al. [56], in which fullness and satiety were significantly increased and hunger significantly decreased after an intake of 100 mg EGCG daily for two months. 

Only one study found an increasing effect on hunger [49].

#### 3.3.2. *Capsicum annuum*

Capsaicinoids (e.g., capsaicin or dihydrocapsaicin) are the bioactive compounds of the chili pepper fruit, genus *Capsicum*, that are responsible for their pungent sensation [20,81,82]. A few studies report that orally ingested capsaicinoids are able to increase satiety [83,84,85]. However, the findings of this review regarding the efficacy of capsaicinoids are not entirely consistent. 

In one of the short-term studies that recorded the outcomes over several hours on one or two days, Hochkogler et al. [57] found that the ingestion of 0.15 mg nonivamide (a capsaicin analog) decreased subjective feelings of hunger over a time of two hours. Appetite-reducing effects were also found by Janssens et al. [21], using 3090 mg red chili pepper (*Capsicum annuum* and *Capsicum frutescens*, 39,050 Scoville heat units, 7.68 mg capsaicin). 

The long-term studies investigated the effects of capsaicinoids over a period of three weeks to around three months. Results showing a reduction of appetite were shown by Reinbach et al. [53], using 1530 mg cayenne daily (a variety of *Capsicum annuum*, 40,000 Scoville heat units, content of capsaicin not reported), at least when combined with a positive energy balance. An RCT of Urbina et al. [58] with a twelve-week intake of capsaicinoid (2 or 4 mg daily, Scoville heat units not reported) as well as an RCT from Lejeune et al. [59] with a thirteen-week intake of 135 mg capsaicin daily did not show a significant treatment effect compared to placebo.

There were two trials indicating that a combination of capsaicinoid with green tea (its main compound being epigallocatechin gallate) could be effective in reducing appetite. In the RCT of Reinbach et al. [53], a daily intake of 3.5 dl green tea drink containing 1795.5 mg catechins combined with capsaicin capsules containing 1530 mg cayenne (40,000 Scoville heat units, content of capsaicin not reported) significantly reduced hunger and increased fullness and satiety after a period of three weeks. In their 2013 trial, Rondanelli et al. [64] found an increased feeling of satiety after ingesting a combination of capsaicinoids, epigallocatechin gallate, piperin, and L-carnitine (content not reported more precisely) for eight weeks.

### 3.4. Secondary Outcome

The occurrence of adverse events was low. Out of 32 analyzed studies, 16 (50%) did not report adverse events at all. In 10 of the remaining 16 trials, none of the tested plant extracts caused an adverse event [15,48,49,54,56,64,68,70,71,76]. Six studies [51,58,63,66,67,74] reported mild adverse events such as gastrointestinal distress (e.g., dyspepsia, flatulence, diarrhea, or constipation), skin rash, or headache. No serious adverse events occurred.

In an RCT of Urbina et al. [58], 23% of the participants in the high-dose capsaicinoid group reported gastrointestinal distress (not described in detail), whereas no participant in the low-dose nor in the placebo group reported such side effects. These findings are in line with recent data suggesting that capsaicin promotes satiety by increasing gastrointestinal distress and bloating sensation [20].

According to Gout et al. [67] the supplementation with Satiereal^®^ caused adverse events like nausea, diarrhea, or reflux in 16% of the trial members, whereas none occurred in another Satiereal^®^ study [68].

Two other trials also reported mild adverse events, which did not differ between the intervention group (Meratrim^®^ and caralluma extract, respectively) and the placebo group [66,74].

### 3.5. Nutritional Regimens

In 5 out of 18 trials recording the outcomes over multiple days, the participants were not requested to change their dietary habits [15,58,67,68,71]. Others had to follow a nutritional regimen, cutting down their daily energy intake by around 2092 kJ (500 kcal) to 3344 kJ (799 kcal) [55,56,64,72] or adhering to a maximal daily intake of around 8400 kJ (2007 kcal) [48,74]. Most diets consisted of 58% to 65% carbohydrates, 15% to 30% fats, and 15% to 25% proteins [48,49,56,63,64,74]. A different regimen was used in a study from Westerterp-Plantenga et al. [54]. The subjects had to follow a protein-enriched diet (46% carbohydrates, 47% proteins, 6% fat) to lose 4 kg body weight before the supplementation period started.

In multiple studies that recorded the outcomes over several hours on just one or two days, respectively, participants were instructed to refrain from exercise 24 to 48 h before test day [28,51,60,61,62,65,75] and to arrive after an overnight fast at the test center [52,57,62,69,70,76,86]. The bioactive phytochemical was mostly taken with a standardized meal, varying from a breakfast meal with approximately 1150 kJ (275 kcal) [73] to a breakfast buffet with a total amount of approximately 32,962 kJ (7878 kcal) [61]. Around 50% to 70% of calories were taken in as carbohydrates, 14% to 31% as fats, and 10% to 20% as proteins.

### 3.6. Quality Assessment of the Studies

The risk of biases of all included studies according to the SIGN checklist [46] is presented in Appendix A.

Several methodological weaknesses were observed, including the inclusion of just one gender [15,28,49,50,57,60,67,71,75,76], single blinding [21,60,70], a high dropout rate (>20%) [21,28,55,58,62], a small group size (parallel-designed RCTs with just 20 participants) [68,87], and a short intervention period (parallel-designed RCTs less than 1 month) [68]. 

Out of the 32 studies, 18 did not report which method they used to randomize the assignment of subjects to the treatment groups (randomization method) and 25 did not report which method they used to ensure that researchers were unaware of the group to which patients were being allocated (concealment method).

## 4. Discussion

### 4.1. Main Findings

This systematic review aimed to present the efficacy and safety of bioactive phytochemicals with appetite/hunger-suppressing and/or satiety/fullness-increasing properties. The findings from 32 published parallel- or crossover-designed RCTs revealed mostly inconclusive evidence that bioactive phytochemicals are effective in suppressing appetite/hunger and/or increasing satiety/fullness. None of the plant extracts tested in several trials showed a consistent positive treatment effect. The occurrence of mild adverse events was low. No serious adverse events were reported.

### 4.2. Comparison to Existing Literature

In a systematic review prepared by Astell et al. [8] from 2013 investigating plant extracts with appetite-suppressing properties on body weight control, 14 studies using 10 different plant extracts (alone or in combination) met their inclusion criteria. The most investigated was *Garcinia cambogia* (alone or combination with another substance) with five RCTs. Only one of those studies revealed a statistically significant positive effect. In that study, Preuss et al. [88] tested (–)-hydroxycitric acid, an active ingredient derived from *Garcinia cambogia*, resulting in a statistically significant reduction of appetite and body weight. In our systematic review, this study was excluded due to toxicological concern with *Garcinia cambogia* [89].

Astell et al. [8] presented three trials comparing changes in body weight, energy intake, fat intake, and hunger and/or satiety between *Camellia sinensis* (green tea) and a placebo group. The average sample size was 150 (range 104 to 240) and the average duration of the studies was 12 weeks (range 12 to 13 weeks). None of them found a significant difference between the two groups [90,91,92]. In our review, only four (44%) out of nine studies that investigated the appetite/hunger-suppressing and/or satiety/fullness-increasing efficacy of *Camellia sinensis* (green tea) revealed a statistically significant positive effect compared to placebo [51,52,53,56].

Other single substances tested by Astell et al. [8] were *Amorphophallus konjac* (no significant effect on energy intake or body weight) [93], *Caralluma adscendens* var. *fimbriata* (significant reduction of hunger levels compared to placebo) [66], *Irvingia gabonensis* (no significant effect on energy intake) [94], fenugreek or *Trigonella foenum-graecum* (no significant effect on appetite, satiety, energy intake, or body weight) [95], *Hoodia gordonii* (no significant effect on satiety, energy intake, or body weight) [96], and *Phaseolus vulgaris* (no significant effect on energy intake, appetite control, hunger, body weight, or body fat) [97]. Tested combinations were *Garcinia cambogia* with *Gymnema sylvestre* (see above) [88], *Garcinia cambogia* with *Amorphophallus konjac* (no significant effect on body weight and energy intake) [98], and *Garcinia cambogia* with green tea extract, *Ephedra sinica*, and *Gymnema sylvestre* (no significant effect on energy intake but significant effect on weight loss) [99].

### 4.3. Definition of Appetite, Hunger, Satiety, and Fullness

It is not clear if the terms appetite, hunger, satiety, and fullness had exactly the same meaning in all studies. In some cases, the terms were analyzed as four different outcomes, in other cases they were used synonymously. Clear and consistent definitions of these terms are necessary.

### 4.4. Instruments to Assess Appetite, Hunger, Satiety, and Fullness

Since appetite, hunger, satiety, and fullness are subjective parameters, it is not easy to quantify them in a valid and reliable way. Most instruments use a questionnaire that is completed before and after consumption of the substance tested, and then again at regular time intervals (e.g., every 30 min) for a couple of hours or until the next meal [100]. Questions used are, for example, “how hungry are you?”, “how full are you?”, “how satiated are you?”. Answers are given as a value in a range, with endpoints defined with terms like “not at all” to “extremely” and often rated on line scales like the visual analogue scale (VAS) from “0 to 100” [100,101]. One criticism of line scales is that the distances between the units do not reflect perceptual distances. For example, on a VAS scale of 0 to 100 for hunger, does a response of 80 indicate twice as much hunger as a response of 40 [100,102]? Another problem is that subjects commonly do not use the full range of the scale and try to avoid extreme responses [102]. 

A validated instrument should have a high reliability (similar results when measurements are repeated) and a high validity (results corresponding accurately to reality) with other instruments. In the case of subjective parameters, it can be difficult to establish these properties since corresponding objective measures are lacking [103]. There are only a few direct comparisons of different scaling methods measuring appetite [100]. Nevertheless, the commonly used line scales seem to have a high reliability, at least regarding the group mean [100,103]. Values at single time points (e.g., baseline, follow-up 1, follow-up 2) have lower reliability than averaged values [100]. Hunger increases and decreases during the day—independent of time since the last meal and time since waking up—following an endogenous circadian rhythm [104]; therefore, the time point of assessment is relevant.

Assessing the validity of such measures is made more difficult since it is not clear if every subject has the same perception of hunger (“is your hunger the same as my hunger”) and if hunger scales really are measuring hunger [100].

### 4.5. Mechanism of Action of Bioactive Phytochemicals

The mechanisms of action behind the appetite-suppressing properties of bioactive phytochemicals suspected to exert such activity are still not completely understood [19,20,21]. Possible targets are, among others, gut-derived hormones (such as GLP-1, PYY, or ghrelin) [20,22,105] or receptors of the neuronal system (such as TRPV1) [23]. The following two chapters will focus on potential mechanisms of *Camellia sinensis* (green tea) and *Capsicum annuum*, the two plants most examined in this systematic review.

#### 4.5.1. *Camellia sinensis* (Green Tea)

The available literature offers limited information on the mechanisms with which *Camellia sinensis* (green tea) and its active compounds could influence appetite. A possible mode of action is the inhibition of the catechol-O-methyltransferase (COMT) through catechins, the enzyme that degrades norepinephrine [79]. Higher concentration or prolonged action of norepinephrine leads to augmentation of thermogenesis and fat oxidation and may also alter appetite [52,77].

According to Fernandes et al. [51], EGCG is able to delay gastric emptying to a small but statistically significant amount, which could also be a possible mechanism of influencing appetite. Josic et al. [52] assumes the opposite. Reduction of gastric emptying usually reduces postprandial glucose concentrations. Since green tea failed to reduce the postprandial concentrations of glucose, reduction of gastric emptying does not seem to be the mechanism of action behind decreasing appetite. Further studies are necessary to elucidate the reason of the observed effects.

#### 4.5.2. *Capsicum annuum*

The appetite-suppressing mechanisms of action of capsaicinoids (derived from the chili pepper fruit, genus *Capsicum*) are not completely understood [20,21].

In a study of Smeets and Westerterp-Plantenga [22], a lunch supplemented with red pepper (cayenne) increased the concentrations of glucagon-like peptide-1 (GLP-1) and decreased the concentrations of ghrelin. GLP-1, produced in the ileum and colon, appears to be a regulator of appetite and food intake and to act as an anorexigenic hormone (low during fasting, increased in response to food intake) [39,106]. On the other hand, ghrelin, predominantly produced in the stomach, seems to stimulate hunger and weight gain and therefore to act as an orexigenic hormone (high during fasting, decreased in response to food intake) [39,107]. The results of a study from Rigamonti et al. [108] are contradictory to this possible mechanism of action of capsaicinoids. No differences in postprandial circulating concentrations of ghrelin and GLP-1 were observed after ingestion of capsaicin or placebo.

Capsaicinoids may also act as appetite suppressants by interfering with the sympathetic nervous system (SNS). They have been reported to enhance catecholamine secretion from the adrenal medulla in rats [109]. This mechanism is caused through activation of the transient receptor potential vanilloid subfamily 1 receptor (TRPV1) [110]. The reaction with TRPV1 also leads to stimulation of peripheral nociceptive neurons, which is responsible for the perception of heat induced by capsaicinoids [23,111]. The release of catecholamines causes a reduction in appetite and is one target of appetite suppressant drugs [112,113].

### 4.6. Implications for Clinical Practice

Due to the heterogeneity of the results of the included studies, no valid quantitative information can be provided to physicians. There were several studies showing a positive treatment effect of a bioactive phytochemical in suppressing appetite/hunger and/or increasing satiety/fullness. Nevertheless, the outcomes remain contradictory.

Although only a small number of adverse events occurred in the analyzed trials and they were all rated as mild, it is important to keep possible toxicity of the bioactive phytochemicals (e.g., *Garcinia cambogia* [89]) in mind when prescribing them to patients. 

### 4.7. Implications for Further Research

Non-uniform usage of the terms appetite, hunger, satiety, and fullness leads to difficulty in interpreting and comparing the results of studies. Clear and consistent definitions of these terms are necessary.

Our systematic review showed contradictory results regarding the appetite/hunger-suppressing and/or satiety/fullness-increasing efficacy of bioactive phytochemicals. Differing dosages and inconsistent declaration of the bioactive phytochemical content (e.g., different catechins) complicated the comparability of the studies. Further research of bioactive phytochemicals with standardized quality is needed to determine the efficacy, optimal dosage, and safety. Studies, where the test agent was not defined and their quality could not be linked to evidence, had to be excluded from the review. Some studies showed significant effects of the bioactive phytochemicals used, but they were often short-term trials. Therefore, the ingestion for longer periods should be investigated. Furthermore, studies investigating plant extracts on an empty stomach should be conducted, to assess the appetite/hunger-suppressing and/or satiety/fullness-increasing efficacy independent of the dietary intake (e.g., different diets with different macronutrient content). Next to oral ingestion of the bioactive phytochemicals, other routes of application (e.g., intranasal or dermal) should be tested. Above all, there is a lack of research investigating the mechanisms of bioactive phytochemicals acting as appetite suppressants.

To increase the internal validity of the studies, future systematic reviews should focus on randomized trials in line with an approved checklist, for example, the Scottish Intercollegiate Guidelines Network (SIGN) checklist [46].

Furthermore, more similar study conditions, especially regarding the time points of follow-ups, and a more consistent and widespread use of instruments could improve the ability of comparing trials focusing on appetite.

### 4.8. Strengths and Limitations

One strength of our systematic review is the use of a structured study protocol, the PRISMA guidelines [35] and the SIGN checklist [46]. We utilized them to guide our search strategy, study selection, extraction of data, and statistical analysis. We thoroughly searched five electronic databases and screened a large number of studies. Moreover, we focused on randomized controlled trials (RCTs).

The main limiting factor of this systematic review is the heterogeneity of the included studies in measuring and reporting appetite, hunger, satiety, and/or fullness. Due to this heterogeneity (different bioactive phytochemicals, appetite scales, follow-up times etc.), we refrained from conducting a meta-analysis. To compensate for this, we aimed for a detailed and qualitative review of the included trials. Inequality of the study conditions and lack of sufficient data prevented us from assessing change in BMI (kg/m^2^), change in waist circumference (cm), or change in hip circumference (cm). In addition, a language bias may have been present, since we restricted our search to English and German language publications. In three studies included in our systematic review, female gender was one of the exclusion criteria [57,60,75]. One important reason to exclude women is to avoid possible effects of the menstrual hormones on food-associated feelings [75]. Janssens et al. [21] solved this problem by conducting the test sessions four weeks apart and therefore ensuring that female participants were in the same phase of their menstrual cycle.

## 5. Conclusions

The findings from 32 published parallel- or crossover-designed RCTs revealed mostly inconclusive evidence that the tested bioactive phytochemicals are effective in suppressing appetite/hunger and/or increasing satiety/fullness. For many of the assessed plant extracts, there were studies that reported strong significant effects, whereas others did not find a clinical effect. There was a high degree of heterogeneity between the included studies that precluded a summary of results by meta-analysis. More systematic and high quality clinical studies are necessary to determine the benefits and safety of phytochemical complementary remedies for dampening the feeling of hunger during dieting.

## Figures and Tables

**Figure 1 nutrients-11-02238-f001:**
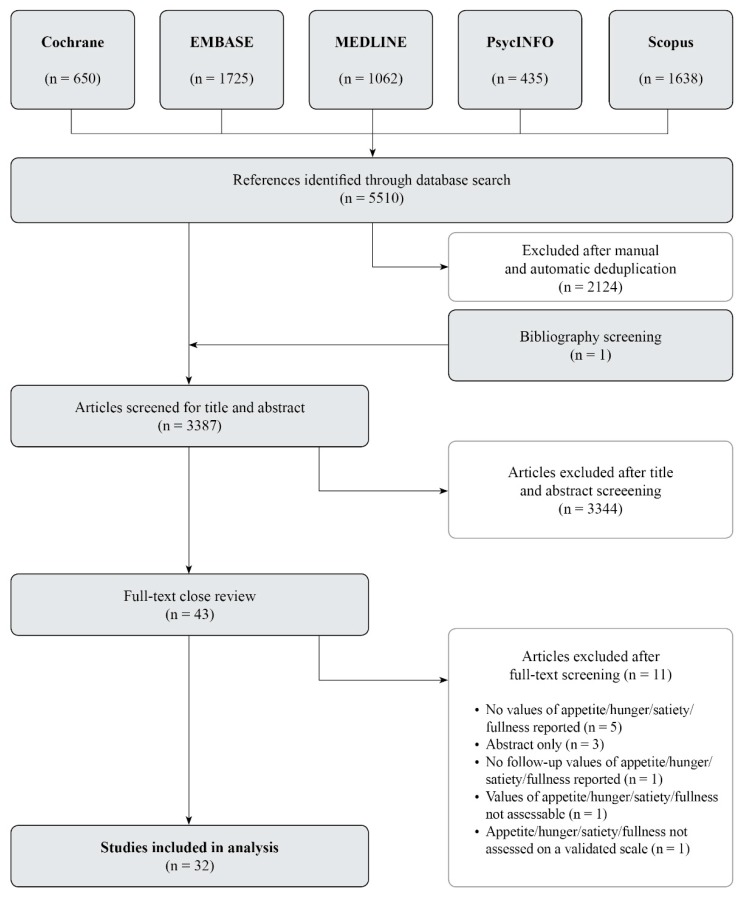
Systematic review flowchart.

**Table 1 nutrients-11-02238-t001:** Types of plant extracts tested in the studies.

Plant Extract(s) of ^1^	Number of Studies	References
*Camellia sinensis* L. Kuntze (green tea)	9	[48,49,50,51,52,53,54,55,56]
*Capsicum annuum* L.	5	[21,53,57,58,59]
*Coffea* sp.	4	[60,61,62,63]
*Camellia sinensis* L. Kuntze (green tea) + *Capsicum annuum* L. + *Piper nigrum* L. + *Fucus vesiculosus* L. ^2^ + *Allium sativa* L.	1	[64]
*Camellia sinensis* L. Kuntze (green tea) + *Ilex paraguariensis* A. St.-Hil. (yerba mate) + *Paullinia cupana* Kunth + *Coffea* sp. + *Serenoa repens* (W. Bartram) Small + *Polygonum multiflorum* Thunb. + *Eleutherococcus senticosus* (Rupr. & Maxim.) Maxim. + *Capsicum annuum* L. + *Pausinystalia yohimbe* Pierre ex Beille ^2^	1	[65]
*Capsicum annuum* L. + *Camellia sinensis* L. Kuntze (green tea)	1	[53]
*Caralluma adscendens* var. *fimbriata* (Wall.) Gravely & Mayur.	1	[66]
*Carum carvi* L.	1	[15]
*Crocus sativus* L.	1	[67]
*Crocus sativus* L. + *Citrus paradisi* Macfad	1	[68]
*Eriodictyon californicum* (Hook. & Arn.) Decne.	1	[69]
*Gentiana lutea* L.	1	[70]
*Hibiscus sabdariffa* L. + *Aloysia citriodora* Palau (syn. *Lippia citriodora)*	1	[71]
*Ilex paraguariensis* A. St.-Hil. (yerba mate)	1	[28]
*Phaseolus vulgaris* L. + *Cynara scolymus* L.	1	[72]
*Salacia chinensis* L.	1	[73]
*Sphaeranthus indicus* L. + *Garcinia mangostana* L.	1	[74]
*Theobroma cacao* L.	1	[75]
*Vitis vinifera* L.	1	[76]

^1^ Accepted scientific names from www.theplantlist.org. For exact composition, see Appendix A. ^2^ Yet unresolved in the plant list (www.theplantlist.org).

**Table 2 nutrients-11-02238-t002:** Primary outcomes for different plant extracts. ^1^

Plant Extract of ^2^	Study	Daily Dosage of Bioactive Phytochemical	Primary Outcomes	Follow-Up
*Camellia sinensis* L. Kuntze (green tea)	Auvichayapat et al., 2008 [48]	100.7 mg EGCG	0	3 months
	Diepvens et al., 2005 [49]	1207 mg catechins (596 mg EGCG, 126 mg EC)	0	3 months
	Dostal et al., 2017 [50]	1315 mg catechins (843 mg EGCG)	0	1 year
	Fernandes et al., 2018 [51]	752 mg EGCG	+	1 day
	Josic et al., 2010 [52]	32.4 mg EGCG 25.5 mg EC	+	1 day
	Mangine et al., 2012 [55]	105 mg EGCG	0	2 months
	Reinbach et al., 2009 [53]	1796 mg catechins	+	3 weeks
	Rondanelli et al., 2009 [56]	100 mg EGCG	+	2 months
	Westerterp-Plantenga et al., 2005 [54]	270 mg EGCG	0	3 months
*Capsicum annuum* L.	Hochkogler et al., 2014 [57]	0.15 mg nonivamide	+	1 day
	Janssens et al., 2014 [21]	7.68 mg capsaicin	+	2 days
	Lejeune et al., 2003 [59]	135 mg capsaicin	0	3 months
	Reinbach et al., 2009 [53]	1530 mg cayenne	+	3 weeks
	Urbina et al., 2017 [58]	2 or 4 mg capsaicinoids	0	3 months
*Coffea* sp.	Greenberg and Geliebter, 2012 [60]	6 mg per kg body weight	0	1 day
	Panek-Shirley et al., 2018 [61]	1 or 3 mg per kg body weight	0	1 day
	Roshan et al., 2018 [63]	800 mg decaffeinated green coffee bean extracts	+	2 months
	Schubert et al., 2014 [62]	4 mg per kg body weight	0	1 day

“0” No statistically significant difference (*p* > 0.05) between intervention and placebo. “+” Statistically significant difference (*p* < 0.05) between intervention and placebo in at least one of the four outcomes (appetite, hunger, fullness, or satiety). ^1^ EC, epicatechin; EGCG, epigallocatechin gallate. ^2^ For exact composition, see Appendix A.

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
