# Peer review of "Appetite-Suppressing and Satiety-Increasing Bioactive Phytochemicals: A Systematic Review"

_nutrients, 2019, doi:10.3390/nu11092238_

Round 1
Reviewer 1 Report
The authors need to include the justification of 20 subjects as the minimal sample size in the manuscript, not just in the response to reviewer comments
Green tea is not an herbal tea, so the author’s response does not make sense on this issue. Typically, “herbal tea” is a term used to describe teas brewed from plants other than Camellia sinensis
There is debate in the literature about whether caffeine, coffee and tea are addictive. I would add a statement stating that these were included because there is debate. Simply stating that they are not addictive is not in agreement with the current views in the literature.
Line 14-15
Improve the clarity of this statement by modifying it as follows “phytochemicals in food supplements are a trending approach to facilitate dieting and to improve patients’ adherence to reduce food and caloric intake.”
Line 31
I think the term “undermine” is not used correctly
Line 33
Dieting (Line 14) and therapeutic fasting are not the same thing. Therapeutic fasting is a subset of diet approaches. Why confuse the two? Be consistent in your terminology
Line 71
The terms isolates seems redundant with the terms extracts and fractions
Line 75
Plant-drug interactions are not referred to as “pharmacovigilance”…it is unclear where the authors are getting these terms and definitions
Line 98
The phrase “non-nutrients in the sense of nutrition needs” is redundant.
Line 143-148
Clearly explain the difference between high ratings (majority of criteria met) vs. acceptable (most criteria met). It is unclear how the authors defined “majority” vs “most” vs “almost none”. These are very subjective terms and do not really inform the reader about how decisions were made
Line 156-158
The primary outcomes should be objective measures of appetite, hunger, satiety and fullness, regardless of the direction of the effect…as stated, it looks like the authors only looked for outcomes in a certain direction: “The primary outcome of interest of this systematic review was the decrease of appetite or hunger, and/or the increase of satiety or fullness”.
Table 1
It is unclear why several plants are listed multiple times, especially with the edits over the original paper….does this indicate mixtures?
Tables
The tables in the edited version are nearly impossible to read. Foor the next revision, please provide a “clean” version as well
Line 381
Why were these studies with Garcinia and other phytochemicals and plants not included in the present review?
Author Response
Comments to the reviews on the manuscript
Appetite suppressing and satiety increasing bioactive phytochemicals: A systematic review
September 5th, 2019
We thank the reviewers and the editor for their very valuable thoughts and comments. We have included the suggestions and revised and improved our manuscript accordingly and now believe to provide balanced clinically relevant information for clinicians and researchers. You find our detailed reply to the points raised by the referees below.
Kind regards
The authors of manuscript 585477 Stuby et al.
Reply to Reviewer #1:
Query |
Reviewer’s comment |
Our answer |
1 |
The authors need to include the justification of 20 subjects as the minimal sample size in the manuscript, not just in the response to reviewer comments |
Thank you for your comment. We added our justification in the manuscript (line 100-102). “This criterion is based on the smallest sample size that was justified by a power analysis from literature previously known to us.” |
2 |
Green tea is not an herbal tea, so the author’s response does not make sense on this issue. Typically, “herbal tea” is a term used to describe teas brewed from plants other than Camellia sinensis |
Thank you for raising this point, which might not be totally clear in the scientific community in the area of food and nutrition. In the field of pharmaceutical application, Green tea is monographed in the European Pharmacopeia as a traditional herbal drug raw material. In addition, the Herbal Medicinal Product Committee (HMPC) of the European Medicine Agency published a monograph on the evidence of green tea as a potential phytotherapeutic agent with legal status of “traditional use”. We included this public available clear opinion from this expert committee as a citation and would like to hint to section 3 Pharmaceutical dosage form” of this official document stating the following wording: “Herbal substance or comminuted herbal substance as herbal tea for oral use.” (Herbal Medicinal Products Committee. Community herbal monograph on Camellia sinensis (L.) Kuntze, non fermentatum folium. EMA/HMPC/283630/2012, London, 2013.) Therefore, green tea is a phytotherapeutically accepted herbal tea, which can also be applied as a food supplement. We added the citations to make the point more clear in line 113/114. |
3 |
There is debate in the literature about whether caffeine, coffee and tea are addictive. I would add a statement stating that these were included because there is debate. Simply stating that they are not addictive is not in agreement with the current views in the literature. |
Thank you for your input. We changed the text accordingly and added some more references (line 111-113): “[…] or if it had an addictive potential (e.g., nicotine or marihuana), We included studies investigating caffeine because there is an ongoing debate in the literature whether caffeine has an addictive potential [40-43]. |
4 |
Line 14-15
Improve the clarity of this statement by modifying it as follows “phytochemicals in food supplements are a trending approach to facilitate dieting and to improve patients’ adherence to reduce food and caloric intake.” |
Thank you for input. We changed the sentence accordingly (Line 13-14). |
5 |
Line 31
I think the term “undermine” is not used correctly |
Thank you for your important comment. We changed the sentence as follows (line 25-28): “More systematic and high quality clinical studies are necessary to determine the benefits and safety of phytochemical complementary remedies for dampening the feeling of hunger during dieting.” |
6 |
Line 33
Dieting (Line 14) and therapeutic fasting are not the same thing. Therapeutic fasting is a subset of diet approaches. Why confuse the two? Be consistent in your terminology |
Thank you for your important input. We changed “therapeutic fasting” to “dieting” (Line 27 and 463) and deleted on line 94 the term. |
7 |
Line 71
The terms isolates seems redundant with the terms extracts and fractions |
Thank you for raising this point on definition of terms. An isolate from a complex herbal extract. |
8 |
Line 75
Plant-drug interactions are not referred to as “pharmacovigilance”…it is unclear where the authors are getting these terms and definitions |
We clarified the sentence (line 65-69) and would like to refer to the reviewer with his/her concern to the following reference: «Adverse drug reactions, including those resulting from interactions between herbal medicines and conventional drugs, are a public health problem worldwide. The need for pharmacovigilance for herb-drug interactions (HDIs) is essential for the identification and assessment of risks of using herbal products (questionable safety, efficacy and quality)» |
9 |
Line 98
The phrase “non-nutrients in the sense of nutrition needs” is redundant. |
We changed the text accordingly (line 89) and deleted “in the sense of nutrition needs”. |
10 |
Line 143-148
Clearly explain the difference between high ratings (majority of criteria met) vs. acceptable (most criteria met). It is unclear how the authors defined “majority” vs “most” vs “almost none”. These are very subjective terms and do not really inform the reader about how decisions were made |
Thank you for your comment. We assessed the methodological quality of the studies according to the SIGN checklist and their proposed rating. Unfortunately, the SIGN does not provide detailed instructions for the final rating of the overall methodological quality. Consequently, the final decision is quite subjective. We rated a study with “++” if at most one criterion was answered with a “no” or “can’t say”, with “+” if at most two criteria were answered with a “no” or “can’t say”, and with “0” if at most four criteria were answered with a “no” or “can’t say”. For Co-RCTs criterion 1.5 is not applicable, however, according to the SIGN list, you can’t choose “NA” as an option, therefore we filled in “can’t say”, but we did not consider it for the final rating. We added this description to the manuscript. (line 132-140). Unfortunately, there was a transcription error while copying the final rating from our excel sheet to the word table. We adjusted Table S7 accordingly. |
11 |
Line 156-158
The primary outcomes should be objective measures of appetite, hunger, satiety and fullness, regardless of the direction of the effect…as stated, it looks like the authors only looked for outcomes in a certain direction: “The primary outcome of interest of this systematic review was the decrease of appetite or hunger, and/or the increase of satiety or fullness”. |
The aim of our review was to investigate bioactive phytochemicals with appetite/hunger suppressing and/or satiety/fullness increasing properties and the systematic literature search was conducted according to this. Consequently, we did not further analyze the potential opposite effects. Nevertheless, this is a very interesting topic and would be worth an own review. In table S6, we analyzed the effects of the included bioactive phytochemicals and only one study (Diepvens 2005) found an opposite effect of Camellia sinensis L. (green tea) on hunger. We added this to the results section (line 236). |
12 |
Table 1
It is unclear why several plants are listed multiple times, especially with the edits over the original paper….does this indicate mixtures? |
Thank you for your comment. Yes, in Table 1 several plants are listed multiple times because they were also tested in combinations with other plants and not only alone. We adjusted the text in the manuscript accordingly (line 182-184). |
13 |
Tables
The tables in the edited version are nearly impossible to read. Foor the next revision, please provide a “clean” version as well |
The tables were edited by the editorial office. We accepted their changes and the tables should be now presented in a “clean” way. |
14 |
Line 381
Why were these studies with Garcinia and other phytochemicals and plants not included in the present review? |
Thank you for your comment. As we stated on line 319-320 we excluded Garcinia combogia due to toxicological concerns according to the recent study of Crescioli et al [83]. In studies, where it was unclear, how the bioactive chemicals were produced from the originating plant and what the content of important marker compounds was, the results can probably not be reproduced and equivalence to a remedy for future application is not given. Evidence is product specific and requires a proper characterization of the agent applied. We included the following sentences line 421-423: “Studies, where the test agent was not defined and their quality could not be linked to evi-dence, had to be excluded from the review.” |

Reviewer 2 Report
A very comprehensive review.
Author Response
Comments to the reviews on the manuscript
Appetite suppressing and satiety increasing bioactive phytochemicals: A systematic review
September 5th, 2019
We thank the reviewers and the editor for their very valuable thoughts and comments. We have included the suggestions and revised and improved our manuscript accordingly and now believe to provide balanced clinically relevant information for clinicians and researchers. You find our detailed reply to the points raised by the referees below.
Kind regards
The authors of manuscript 585477 Stuby et al.
Reply to Reviewer #2:
Query |
Editors’s comment |
Our answer |
1 |
A very comprehensive review. |
Thank you very much for your kind statement. It is very much appreciated. |

Round 2
Reviewer 1 Report
Authors have adequately addressed my concerns
This manuscript is a resubmission of an earlier submission. The following is a list of the peer review reports and author responses from that submission.
Round 1
Reviewer 1 Report
Introduction: it may be beneficial to give a brief overview of the mechanisms by which phytochemicals may influence hunger and satiety for improved context
Line 40: I believe the word “ultima” is meant to be ultimate
Line 49
The term phytonutrients is incorrect. Phytonutrients are plant compounds that are essential nutrients for humans, which does not apply to the compounds discussed here. Phytonutrients include vitamin C, pro-Vitamin A carotenoids, vitamin E compounds, etc. Better terms would be phytochemicals, plant natural products or plant secondary metabolites.
Line 68-73
Some terms that probably should have been used are “phytochemical”, “natural product”, major groups of phytochemicals such as “phenol”, “polyphenol”, “terpene”, ‘terpenoid”, “alkaloid”, as well as specific compounds or plants that are known to affect appetite. It seems that some studies were likely missed because of this. For researchers who regularly work with phytochemicals, the search terms used by the authors are not regularly used in journal articles because they are so generic. If I were writing I paper on the influence of the compounds I study (polyphenols) on appetite, I would never include any of the terms the authors used since they are so generic…these terms are more typical in commercial marketing than in scientific papers. I find it very odd that these were the terms used and indicative of a lack of expertise on plant bioactives on the part of the research team. This is a real concern in terms of whether all relevant papers were actually located.
Line 78
Please justify and explain the selection of 20 participants as the minimum. This seems somewhat random
Line 78-80
This sentence doesn’t make any sense: “In case of crossover RCTs, each test person underwent both intervention as well as control treatment. Therefore, the total number of participants could be smaller, as long as the substances were tested on more than 20 participants”. It is unclear how having 20 people in a crossover trial could result in a sample size less than your minimum of 20.
Line 82-83
Why were subjective measures of satiety used (VAS) but objective endocrine markers of satiety were not considered?
Line 86
Why was caffeine included when the authors explicitly state that they excluded addictive compounds?
Line 103
The authors state “if further research was unlikely to change the results, the study assessed a high rating”. It is unclear how the authors could possibly know or even guess that. Odd study results happen all the time. Based on subsequent lines it appears they think that good design leads to the same results all or even most of the time. That’s simply not true.
Line 149
Technically green tea doesn’t meet the inclusion criteria (it is a conventional foodstuff, it is not a plant extract or an isolated compound). The criteria seem very haphazardly applied.
Table 1
Sodium alginate, calcium, linoleic acid, sodium phosphate, multivitamin/mineral and galactomannan seem out of place with the others. It really isn’t clear what types of compounds the authors were trying to study. In particular, calcium, linoleic acid, sodium phosphate, and multivitamin/mineral have no clear relationship specifically to plants.
Table 2 should include doses
Section 3 is highly superficial and barely worthy of being called a review
Line 374-375
It is unclear how the last sentence of the paper (Nevertheless, we recognize that phytonutrients have promising potential for the control of appetite and satiety without causing serious adverse events) can be supported in line of their previous statement in lines 262-265 (The findings from 57 published parallel- or crossover-designed RCTs revealed mostly inconclusive evidence that plant extracts are effective in suppressing appetite/hunger and/or increasing satiety/fullness. None of the substances tested in at least three trials showed a consistent positive treatment effect)
Figure 1: Consider making the figure larger or changing the font, it is difficult to read as it is
Did the authors make note of differences in diet between these studies? How were differences accounted for in the current review? What about differences in phytochemical dosage? Similarly, it appears that the majority of the studies appear to be acute, with some showing significant treatment effects in as little as one day. The authors may wish to elaborate on the lack of long term studies or acknowledge it as a limitation
Reviewer 2 Report
There are some issues that need to be revised or explained in this study.
1- Since this is a systematic review and the main work done was statistics, there is no enough information about running of stats in this study. Which software did you use? Which test did you run? You need to bring all the information.
2- Out of my curiosity, did you see in opposite effect of phytonutrients on satiety and hunger? Did they increase hunger or decrease satiety? It was worth to discuss this effects of phytonutrients as well.
3- Fig 1 has a very low quality and it is not readable.